# What are the Essential Factors in Crafting Effective Long Context Multi-Hop Instruction Datasets? Insights and Best Practices

## Abstract

Recent advancements in large language models (LLMs) with extended context windows have significantly improved tasks such as information extraction, question answering, and complex planning scenarios. In order to achieve success in long-context tasks, a large amount of work has been done to enhance the long-context capabilities of the model through synthetic data. Existing methods typically utilize the Self-Instruct framework to generate instruction-tuning data for better long-context capability improvement. However, our preliminary experiments indicate that less than 35% of samples generated by Qwen-$2_{72B}$ are multi-hop, and more than 40% exhibit poor quality, limiting comprehensive understanding and further research. To improve the quality of synthetic data, we propose the Multi-agent Interactive Multi-hop Generation (`MIMG`) framework, incorporating a Quality Verification Agent, a Single-hop Question Generation Agent, a Multiple Question Sampling Strategy, and a Multi-hop Question Merger Agent. This framework improves the data quality, with the proportion of high-quality, multi-hop, and diverse data exceeding 85%. Furthermore, we systematically investigate strategies for document selection, question merging, and validation techniques through extensive experiments across various models. Our findings show that our synthetic high-quality long-context instruction data significantly enhances model performance, even surpassing models trained on larger amounts of human-annotated data.

## 1 Introduction

Recently, large language models (LLMs) with long-context windows have significantly improved tasks such as information extraction, question answering, and even complex planning scenarios (Liu et al., 2024a; Bai et al., 2024b; Hu et al., 2023; 2024; Xu et al., 2024b). Research on developing long-context LLMs has predominantly focused on extending the context window (Ding et al., 2024; Jin et al., 2024; Peng et al., 2024). Nevertheless, in practical applications, merely expanding the context window is insufficient for effectively utilizing long-context (Hsieh et al., 2024; Huang, 2024), which presses a need for training to optimize utilization of long-context (Zhang et al., 2024), especially in instruction-tuning (IT) (Fu et al., 2024b). In the IT phase, a large amount of high-quality long-context IT data is required. However, acquiring such data is challenging, with annotation costs significantly higher than those for short-context data (Bai et al., 2024b; Xiong et al., 2024). To address this, Xiong et al. (2023) and Bai et al. (2024a) have explored leveraging LLMs to generate IT data using the Self-Instruct framework (Wang et al., 2023b), thereby mitigating the scarcity of long-context IT data.

Moreover, the challenge often lies not in extracting single-hop information, but in integrating multiple hops of data from the long context to reach complex conclusions. Despite this, existing studies struggle to produce high-quality, multi-hop IT data. This gap stems from insufficient attention to the data synthesis process and factors influencing data effectiveness. As illustrated in Figure 1 (a), our preliminary manual annotation experiments show that direct self-instruction yields less than 35% multi-hop samples, with high-quality examples representing only 60%. Additionally, sample diversity remains problematic, with over 45% of the samples exhibiting semantic duplication. These issues hinder comprehensive understanding and further advancement in this domain.

Motivated by these challenges, this paper systematically investigates the research question: *What are the essential factors in crafting effective long-context multi-hop instruction datasets?* To address this, we propose a Multi-agent Interactive Multi-hop Generation (MIMG) framework. First, to ensure the quality of long-context IT data, we introduce a Quality Verification Agent to automatically verify the quality of generated samples during the whole process. Second, to incorporate multi-hop reasoning, we develop a Single-hop Question Generation Agent, followed by a Multi-hop Question Merging Agent for stepwise synthesis of multi-hop problems. Finally, to ensure diversity, we implement multiple question sampling strategies within the Single-hop Question Generation Agent to minimize repetition and promote variety in the multi-hop questions. As shown in Figure 1 (b),

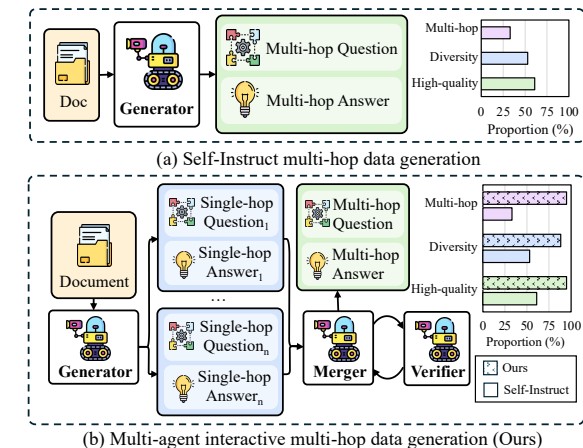

(a) Self-Instruct multi-hop data generation

(b) Multi-agent interactive multi-hop data generation (Ours)

Figure 1: Comparison between traditional self-instruct-based data synthesis method and our Multi-agent Interactive Multi-hop Generation (MIMG) framework, where all data are generated by Qwen-2$_{72B}$ (Yang et al., 2024).

our method significantly improves data quality, with over 85% of the data being multi-hop, high-quality, and non-duplicative.

To optimize long-context instruction data creation, we systematically examine several sub-questions, such as the efficacy of validation techniques, document selection strategies, and the impact of question merging methods. We conduct extensive experiments, applying 17 strategies across 10 domains and 5 models. Our results demonstrate that MIMG significantly enhances data quality. Notably, models trained on the synthetic high-quality data show an average improvement of 7.54%, surpassing models trained on larger human-annotated datasets.

The main contributions of our work are as follows:

- **An Extensive Exploration of Best Practices:** This study examines strategies for generating high-quality multi-hop instructional data, identifying key factors influencing long-context data quality. These include scoring verifiers, question-then-answer generator, question-based sampling, and merging strategies based on question-answer pairs.
- **A Novel Data Synthesis Framework:** We introduce the Multi-Agent Interactive Multi-hop Generation (MIMG) framework, which leverages multiple agents interaction to significantly enhance the quality and relevance of the synthesized data.
- **A Large Long-Context Instruction Dataset for Effectively Enhanced Long-Context Utilization:** Our synthetic dataset, (LongMIT), has shown superior performance across various models. It not only improves long-context utilization but also surpasses larger human-labeled datasets, highlighting its practical contribution to advancing long-context LLMs.

## 2 FRAMEWORK

Our framework consists of four main components: Quality Verification Agent (§ 2.1), Single-hop Question Generation Agent (§ 2.2), Multiple Question Sampling (§ 2.3), and Multi-hop Question Merging Agent (§ 2.4). Specifically, the Quality Verification Agent is first designed as a validator to control and supervise the data quality at each stage. The Single-hop Question Generation Agent then generates simple, direct single-hop questions. Next, Multiple Question Sampling strategies expand on this by sampling questions that cover various documents, enhancing multi-hop instruction generation. Finally, the Multi-hop Question Merging Agent integrates these single-hop questions into coherent multi-hop questions, requiring information synthesis from multiple document parts. The detailed architecture is illustrated in Figure 2.

### 2.1 QUALITY VERIFICATION AGENT

The first module in our framework is Quality Verification Agent, which globally supervises and ensures that the generated samples from each step meet a certain standard of quality. This component involves two main processes:

Figure 2: The overall process of our Multi-agent Interactive Multi-hop Generation (MIMG) data synthesis framework.

**Verification Strategy:** This includes additional heuristic strategies to judge which samples should be contained as high-quality data. Specifically, we utilize two wide-used verification strategies:

- **Scoring:** We prompt LLMs to generate continuous scores, manually set a more reliable threshold score based on the validation set, and set those exceeding the threshold score as high-quality data. Formally, given a sample $s$, we select the high-quality data as follows:

$$\mathcal{V}(s|\mathcal{M}) = \begin{cases} \texttt{Approved} & \texttt{Score}(s|\mathcal{M}) > \theta; \\ \texttt{Rejected} & \texttt{Score}(s|\mathcal{M}) \leq \theta, \end{cases} \tag{1}$$

where $\texttt{Score}(s|\mathcal{M})$ represents the model score of sample $s$ based on model $\mathcal{M}$, and $\theta$ is the threshold.

- **Classification:** We prompt LLMs to generate binary classification and select those classified as high-quality data. Formally, given a sample $s$, we select the high-quality data as follows:

$$\mathcal{V}(s|\mathcal{M}) = \begin{cases} \texttt{Approved} & \texttt{Class}(s|\mathcal{M}) = 1; \\ \texttt{Rejected} & \texttt{Class}(s|\mathcal{M}) = 0, \end{cases} \tag{2}$$

where $\texttt{Class}(s|\mathcal{M})$ represents the binary classification process of sample $s$.

**Verification Condition:** This involves setting specific conditions $\mathcal{C}$ that both questions and answers must meet to be considered high-quality verification ($\mathcal{V}(s|\mathcal{M},\mathcal{C})$). The process includes:

- **Criteria Perspectives:** Criteria include relevance to the document, clarity, factual accuracy, logical coherence, and complexity of the question and answer. Formally, these perspectives can be formulated as:

$$\mathcal{C} = \{c_1, \dots, c_n\}, \tag{3}$$

where $c_i$ denotes the $i$-th criteria instruction. $n$ denotes the number of criteria perspectives.

- **Auxiliary Context Information:** We integrate additional contextual instructions to enhance the model's accuracy and robustness, like guidelines. These conditions are formally represented as:

$$\mathcal{C} = \{c_1, \dots, c_n\} \oplus \texttt{Context}, \tag{4}$$

where the $\texttt{Context}$ denotes the context including auxiliary guidelines.

- **Auxiliary Generation Information:** We enable the model to provide more reasoning rationale during output generation and observe whether this improves the robustness and accuracy of the verification process.

$$\mathcal{C} = \{c_1, \dots, c_n\} \oplus I_R, \tag{5}$$

where the $I_R$ denotes the instruction that can prompt LLM to generate rationales.

## 2.2 SINGLE-HOP QUESTION GENERATION AGENT

This phase generates single-hop questions and answers from individual documents, encompassing the following components:

**Generation Backbone:** This component utilizes a robust LLM to generate valid and relevant single-hop questions and answers from each document. Multiple questions and answers are produced per document to ensure a diverse foundation for multi-hop question development. We thoroughly examine various LLMs, including both open-source and close-source models, across different scales.

**Generation Strategy:** The strategy employs a structured approach to extract potential questions from the text, using the following techniques:

- **Rationale-based Question Generation:** Chain-of-Thought (CoT) prompting (Wei et al., 2022) has been recognized for its role in improving performance on long-text tasks (Li et al., 2024). Building on this, our study investigates whether generating questions from a long document, supported by rationale, can enhance the understanding of the document's inherent reasoning.

- **Question-Answering Generation Order:** Furthermore, we aim to evaluate whether the sequence of generating questions and answers impacts the overall effectiveness. Specifically, generating the question prior to the answer may reduce the reasoning complexity and improve the quality of the model's output compared to a simultaneous generation approach.

## 2.3 MULTIPLE QUESTION SAMPLING

In order to further optimize the diversity of generated samples, we introduce Multiple Question Sampling strategy to create multi-hop questions by sampling and combining questions from multiple questions and documents. It mainly involves the following two strategies:

**Retrieval Strategy:** This strategy identifies relevant questions and documents for multi-hop question creation. Using relevance sampling, a question semantic relevance matrix is generated, assessing the semantic connections between questions across different documents and guiding the sampling process. The strategy includes:

- **Probability-Based Sampling:** This method evaluates document relevance based on the probability and occurrence of specific keywords related to the questions, like BM25 (Robertson et al., 1995; 2009), and LDA (Hoffman et al., 2010).

- **Semantic-Based Sampling:** This approach assesses the relevance by analyzing the semantic similarity between questions and documents, like embedding similarity.

**Sampling Strategy:** Based on the relevance matrix, the most related questions are selected for merging. This involves choosing questions that are both relevant and complementary, ensuring that the resulting multi-hop questions are coherent and contextually rich. The strategy includes:

- **Intra-Document Sampling:** This strategy focuses on selecting questions within the same document to ensure internal coherent multi-hop data.

- **Inter-Document Sampling:** This strategy involves selecting questions from different documents to ensure a broader contextual coverage.

## 2.4 MULTI-HOP QUESTION MERGING AGENT

The final step merges sampled questions into coherent multi-hop questions, involving two modules:

**Merging Backbone:** We utilize LLM to combine the sampled questions and answers into meaningful multi-hop questions and answers. The model leverages context and semantic understanding to ensure that the merged questions are logically consistent and contextually accurate. The backbone includes 5 classic LLMs.

**Merging Strategy:** This includes rules and heuristics to ensure the merged questions are logically consistent and contextually accurate. The strategy includes:

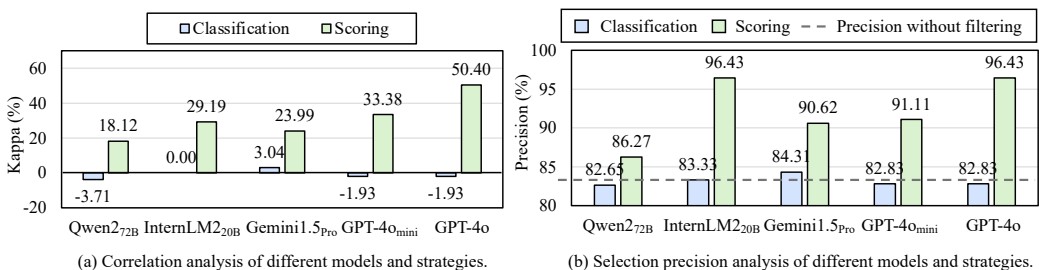

(a) Correlation analysis of different models and strategies.  (b) Selection precision analysis of different models and strategies.

Figure 3: The analysis of different verification strategies in quality verification, where includes 5 models: Qwen2-72B-Instruct (Yang et al., 2024); InternLM2-20B (Cai et al., 2024); Gemini-1.5-Pro (Reid et al., 2024); GPT-4o-mini and GPT-4o (Achiam et al., 2023).

- **Document-Based Merging:** To further reduce input tokens, we explore whether long documents need to be added to large model inputs to enhance merging performance. Formally, the merging process can be represented as:

$$Q_m = \mathcal{M}(Q_1, Q_2, \ldots, Q_n | C), \tag{6}$$

  where $Q_1, Q_2, \ldots, Q_n$ are the sampled single-hop questions, and $Q_m$ represents the merged multi-hop question. $C$ denotes context whether utilize documents.

- **Rationale-Based Merging:** This method leverages the underlying rationale or reasoning behind the original questions to guide their integration, ensuring that the combined question preserves the intended meaning and context of the individual components. Formally, this merging process can be expressed as:

$$R \oplus Q_m = \mathcal{M}(Q_1, Q_2, \ldots, Q_n), \tag{7}$$

  where $R$ represents the rationale or underlying reasoning, and $\oplus$ denotes the connector vocabulary in generated response.

Furthermore, we explore the creation of both intra-document and inter-document multi-hop instruction samples for different scenarios.

## 3 EXPLORATION

This section mainly explores each component of the framework to enhance data quality, including verification strategies and criteria in the Quality Verification Agent (§3.1), generation backbone and strategies in Single-hop Question Generator Agent (§3.2), retrieval and sampling strategies in Multiple Question Sampling (§3.3), and merging backbone and strategies in Multi-hop Question Merging Agent(§3.4).

### 3.1 QUALITY VERIFICATION AGENT

#### 3.1.1 VERIFICATION STRATEGY

Currently, the most widely employed strategies for model verification are scoring and direct classification. We evaluated the consistency and precision of both approaches by comparing them with human annotations in the sample analysis of data generated from long contexts.

**Scoring is a Better Verification Strategy Compared with Classification.** As shown in Figure 3 (a), the scoring strategy shows significantly higher kappa and precision scores compared to binary quality classification. This statistical improvement suggests that scoring better captures the nuances of human judgments. This observation aligns with findings in short-context scenarios (Fu et al., 2024a), reinforcing the generalizability of scoring strategies across different lengths of textual data.

**LLM is not a long-context annotator but a good selector.** As depicted in Figure 3 (a), in contrast to their performance in short-context verification (Wang et al., 2023a; Fu et al., 2024a), LLMs demonstrate minimal agreement with human annotators in long-context scenarios, reflected in low kappa scores. This suggests challenges in maintaining annotation consistency due to the cognitive load and interpretative variations over extensive information.

Despite this, as demonstrated in Figure 3 (b), LLMs consistently achieve nearly perfect precision, indicating robust capability in identifying and selecting relevant data. This distinction underscores the potential of LLMs as effective tools for data filtering and prioritization in long-context environments, contrasting their role as accurate annotators in short-context scenarios.

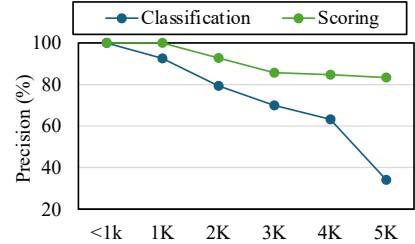

Figure 4: Performance of different models for generating single-hop questions.

**Scoring alleviates the long context bias but classification does not.** We further analyze why classification strategies are less effective in long contexts by examining precision across different context lengths. As shown in Figure 4, the scoring strategy exhibits higher precision and greater robustness in extended contexts than classification, which explains the weaker performance of classification in these settings. Following the conclusions from previous analyses, subsequent experiments will adopt the Scoring strategy. Verifier precision will measure the quality, while data quality will be evaluated by the data retention ratio.

### 3.1.2 VERIFICATION CONDITIONS

To deeply understand what factors affect the verification of long text data quality, we further explored from three perspectives: scoring perspective, guidelines, and whether rationale is included for scoring.

**More scoring perspectives reduce long-context bias.** As illustrated in Figure 5 (a), incorporating more scoring perspectives significantly enhances the accuracy and robustness of filtering long-context data. Unlike short contexts, long contexts introduce noticeable bias in judgments. When fewer than 3 perspectives are used, performance gains are minimal, and the model often overestimates irrelevant samples, leading to poor selection results. However, increasing the number of perspectives markedly improves labeling accuracy, effectively mitigating biases associated with longer contexts. See Appendix A.2.2 for more details.

**Effective verifiers adhere to annotation standards aligned with human judgment.** To assess whether incorporating additional scoring criteria enhances the model's verification performance, we specify the criteria for each score in detail. As illustrated in Figure 5 (b), interestingly, the guideline does not include supplementary information during the annotation process for advanced models. This observation suggests that effective verifiers inherently follow annotation standards that well align with human judgment.

**Incorporating rationale enhances robustness in diverse long contexts.** Our methodology necessitates extension across numerous domains, emphasizing the criticality of robustness across diverse domains. Contextualizing the role of CoT (Wei et al., 2022; Qin et al., 2023), we evaluate model performance across various domains, specifically in wiki-like knowledge and paper analysis domains. As illustrated in Figure 5 (c), incorporating rationale enables the model to maintain high performance across diverse contexts. Without rationale, performance decreases by more than 8.6% when confronted with different domains. Conversely, adding rationale during validation results in minimal performance variation, with fluctuations in precision limited to 1.8% at most.

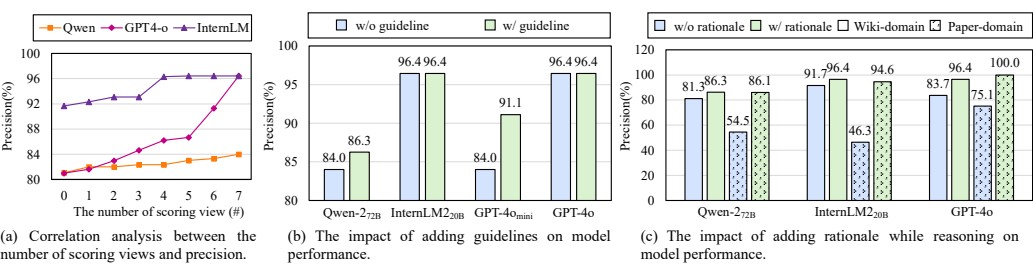

(a) Correlation analysis between the number of scoring views and precision.

(b) The impact of adding guidelines on model performance.

(c) The impact of adding rationale while reasoning on model performance.

Figure 5: The analysis of different verification conditions on quality verification.

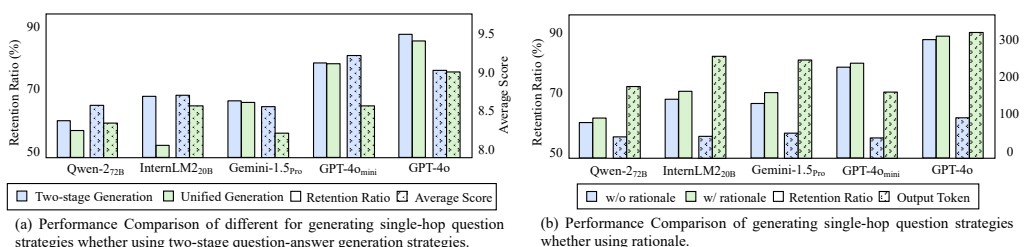

(a) Performance Comparison of different for generating single-hop question strategies whether using two-stage question-answer generation strategies.

(b) Performance Comparison of generating single-hop question strategies whether using rationale.

Figure 7: The analysis of generation strategies in single-hop question generation agent.

## 3.2 SINGLE-HOP QUESTION GENERATION AGENT

### 3.2.1 GENERATION BACKBONE

In practice, effective models must be capable of synthesizing high-quality data. To this end, we explored the suitability of several commonly used LLMs for single-hop data synthesis.

**Open-source LLMs effectively generate single-hop questions.** As shown in Figure 6, smaller open-source LLMs demonstrate high retention rates with cost-efficientiveness, reflecting their capability to understand and generate single-hop questions from a given context.

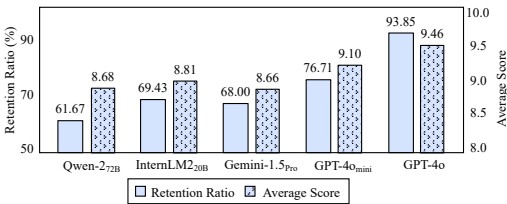

**Stronger LLMs can generate better single-hop question generation but higher cost.** As shown in Figure 6, more advanced LLMs increase data retention and enhance the quality of

Figure 6: Performance of different models for generating single-hop questions.

generated questions. However, these improvements are not cost-proportional, raising concerns about the economic viability of employing stronger models for single-hop question generation.

### 3.2.2 GENERATION STRATEGY

Furthermore, we explore whether employing a question-then-answering approach, supplemented by rationale, enhances the quality of synthetic single-hop questions.

**Question-then-answering works better than generating data from scratch.** To assess whether a single or multiple stage of generation is more effective, we compare two sample generation strategies: unified question-answer and question-then-answer generation. As shown in Figure 7 (a), generating the question before the answer substantially enhances data quality. It improves both the retention rate and the data quality score, especially open-sourced LLMs, confirming its superiority. For more implementation details, see Appendix A.2.3.

**Generating with rationale can improve the generated quality but much higher token cost.** As illustrated in Figure 7 (b), adding rationale makes questions more relevant and insightful with higher quality. However, the improvement brought by the rationale is minimal, while the token consumption triples, making it economically inefficient.

## 3.3 MULTIPLE QUESTION SAMPLING

### 3.3.1 RETRIVAL STRATEGY

This strategy involves identifying relevant documents and constructing a semantic relevance matrix to guide sampling based on both keyword and semantic scoring of documents and questions. Observations on these strategies include:

**Embedding similarity is critical for multi-question sampling.** We assess the effectiveness of various similarity measures by examining three metrics: embedding similarity (using BGE embeddings (Xiao et al., 2023)), BM25, and LDA. As shown in Figure 8 (a), BGE embeddings enable the model to select more relevant questions, enhancing sample quality.

**Question similarity outweighs document similarity.** We also explore which aspects most influence sample quality. Figure 8 (b) demonstrates that question-based sampling significantly outperforms document-based strategies, as questions provide more contextual relevance.

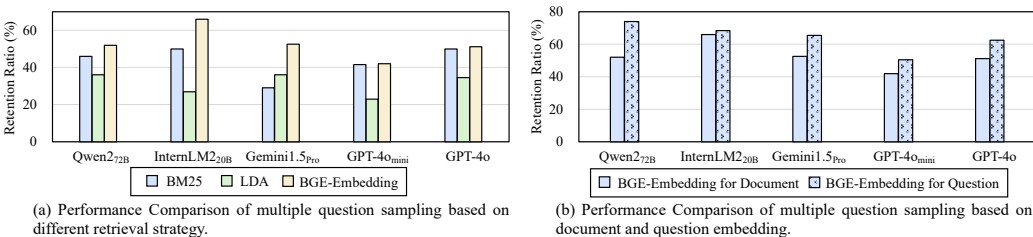

(a) Performance Comparison of multiple question sampling based on different retrieval strategy.

(b) Performance Comparison of multiple question sampling based on document and question embedding.

Figure 8: The analysis of multiple question sampling.

### 3.3.2 SAMPLING STRATEGY

It selects semantically related and complementary questions from within and across documents to form coherent and contextually rich multi-hop questions.

**Intra-Document generates better quality but less diversity.** As shown in Figure 9, sampling questions within the same document results in more coherent and contextually aligned questions. However, this method may limit question diversity since they all stem from the same source.

**Inter-Document generates less quality but more diversity.** As shown in Figure 9, sampling questions from multiple documents introduces a broader range of topics and contexts, enhancing diversity. However, this increased diversity can reduce the coherence and relevance of questions due to larger topic gaps.

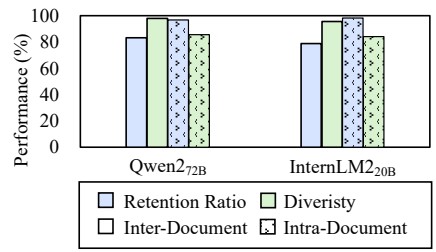

Figure 9: Performance Comparison of multiple question sampling based on different sampling strategies.

### 3.4 MULTI-HOP QUESTION MERGING AGENT

#### 3.4.1 MERGING BACKBONE

We use LLM to merge sampled questions and answers into meaningful multi-hop versions, ensuring logical consistency and contextual accuracy with the help of 5 classic LLMs. The observations are as follows:

**Open-sourced LLMs can well merge multi-hop question generation.** As shown in Figure 10, all models are greatly capable of handling complex question generation tasks that require multiple steps of reasoning or integration of information.

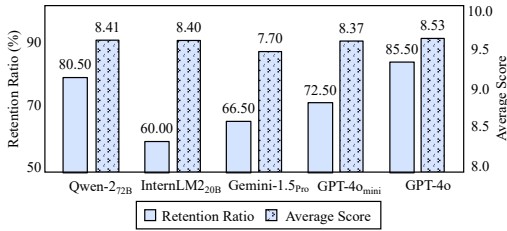

(a) Performance of different models for merging multi-hop questions.

Figure 10: Performance of different models for merging multi-hop questions.

#### 3.4.2 MERGING STRATEGY

**Question-answer pairs are enough for multi-hop instruction merging.** To minimize input tokens, we assess if long documents are necessary for enhancing merging performance. Figure 11 (a) shows

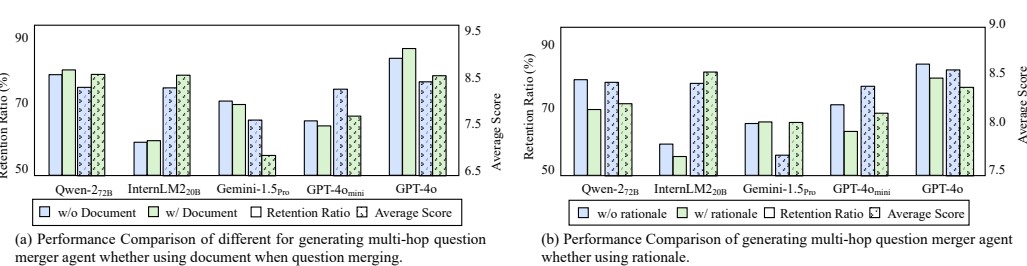

(a) Performance Comparison of different for generating multi-hop question merger agent whether using document when question merging.

(b) Performance Comparison of generating multi-hop question merger agent whether using rationale.

Figure 11: The analysis of multi-hop question merging agent.

that adding documents often fails to consistently improve performance and instead increases input tokens. Thus, simple question-answer pairs effectively achieve multi-hop merging.

**Merging with rationale can not improve the merging quality.** Generally, generating content with rationales can enhance its quality (Qin et al., 2023; 2024). However, as depicted in Figure 11 (b), unlike single-hop generation, rationales in a multi-hop generation do not aid in forming coherent and logical questions. Our quantitative analysis further reveals that large models often misinterpret rationales within queries and merging strategies, leading to frequent CoT failures. Thus, multi-hop synthesis should avoid using additional rationales.

## 4 DATA UTILIZATION

### 4.1 INSTRUCTION DATASET CONSTRUCTION

To expand the domain coverage and handle longer contexts, we extended the instruction fine-tuning data across 9 domains and 2 languages. All base documents were sourced from pre-trained datasets to prevent data leakage. Our Long Multi-hop Instruction-Tuning dataset (`LongMIT`) results in a retention rate of over 90% in GPT-4o verification in 200 sampled samples, confirming the high quality and generalizability of our pipeline. To balance the cost and effectiveness of generating data, `LongMIT` are generated based on Qwen2-72B-Instruct, and verified based on InternLM2-20B. See Appendix A for more details.

### 4.2 DATA SYNTHESIS EFFICIENCY

Given the high cost of data generation, we consider both cost and data quality when synthesizing `LongMIT`. To assess the effectiveness of this balance, we compare the proportion of high-quality data and the token cost for 200 samples generated under different strategies. As shown in Figure 12, strategies with open-source models achieve a high-quality proportion even comparable to the highest quality strategies with GPT4o,

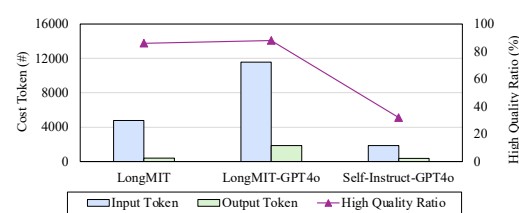

Figure 12: Comparison of the quality and token consumption on different generation strategies.

but at only one-third of the token cost. Furthermore, our approach significantly enhances data quality with minimal additional token expense compared to traditional methods. For more implementation details, see Appendix B.

### 4.3 PREVIOUS INSTRUCTION DATASET

(1) `ChatQA` (Liu et al., 2024b) uses manually annotated long text instruction-following data. (2) `LongAlign` (Bai et al., 2024a) leverages Claude's generative abilities to create 10K QA pairs for training. (3) `LongAlpaca` (Chen et al., 2024b) integrates a large amount of paper QA corpus

| Model | NarrativeQA | 2WikiMQA | DuReader | HotpotQA | MultifieldQA$_{en}$ | MultifieldQA$_{zh}$ | MuSiQue | Qasper | AVG |
|---|---|---|---|---|---|---|---|---|---|
| | | | | InternLM2-1.8B (Cai et al., 2024) | | | | | |
| +ChatQA2 | 18.50 | 35.00 | 29.00 | 46.00 | 64.00 | 58.00 | 19.50 | 38.50 | 38.56 |
| +LongAlign | 25.00 | 33.00 | 25.00 | 49.50 | **76.00** | 67.50 | 24.50 | 44.00 | 43.06 |
| +LongAlpaca | 25.00 | 23.50 | 29.00 | 49.50 | 70.00 | 67.00 | 24.50 | 45.00 | 41.69 |
| +NQ | 17.00 | 25.50 | 33.50 | 35.00 | 60.00 | 67.00 | 14.50 | 44.00 | 37.06 |
| +LongMIT | **26.00** | **35.50** | **60.00** | **56.00** | 75.33 | **75.50** | **29.00** | **47.50** | **50.60** |
| | | | | LLaMA3-8B (Dubey et al., 2024) | | | | | |
| +ChatQA2 | 24.00 | 41.00 | 50.00 | 49.00 | 64.00 | 69.00 | 26.00 | 51.50 | 46.81 |
| +LongAlign | 29.00 | 44.50 | 56.50 | 56.50 | 79.33 | 80.50 | 21.50 | 55.50 | 52.92 |
| +LongAlpaca | 18.00 | 50.00 | 48.00 | 55.50 | 76.67 | 80.00 | 27.50 | **60.50** | 52.02 |
| +NQ | 21.00 | 42.00 | 63.00 | 59.50 | 78.00 | 74.00 | 29.00 | 54.00 | 52.56 |
| +LongMIT | **36.50** | **67.50** | **74.00** | **71.00** | **87.33** | **84.50** | **39.50** | 54.00 | **64.29** |
| | | | | InternLM2-7B (Cai et al., 2024) | | | | | |
| +ChatQA2 | 31.00 | 42.00 | 38.50 | 61.00 | 70.67 | 33.00 | 28.50 | 53.00 | 44.71 |
| +LongAlign | 45.00 | 40.00 | 60.00 | 65.50 | 74.67 | 86.00 | 34.00 | 56.50 | 57.71 |
| +LongAlpaca | 45.00 | 50.50 | 44.00 | 64.50 | 75.33 | 47.50 | 35.50 | 56.50 | 52.35 |
| +NQ | 12.50 | 37.50 | 61.50 | 45.50 | 75.33 | 77.00 | 21.00 | 57.50 | 48.47 |
| +LongMIT | **46.50** | **57.00** | **74.00** | **73.00** | **91.33** | **91.00** | **45.00** | **62.00** | **67.48** |

Table 1: Main accuracy results by evaluation by GPT-4o, where all benchmarks comes from the LongBench (Bai et al., 2024b). More evaluation on Ruler (Hsieh et al., 2024) are shown in Table 2.

with additional short instruction-following examples. (4) `NQ` (Kwiatkowski et al., 2019) is a human-annotated long-context data with a series of natural questions.

### 4.4 The Results of Instruction-Tuning

Based on a substantial volume of synthesized data, we conduct instruction-tuning to further assess its utility. As shown in Table 1, our synthesized data significantly enhances the long-context QA capabilities of various LLMs, achieving an average improvement of at least 7.54% on average. Notably, multi-hop benchmarks like 2WikiMQA (Ho et al., 2020), MuSiQue (Trivedi et al., 2022), and HotpotQA (Yang et al., 2018) show more pronounced improvements. Moreover, as shown in the case study in Appendix D, the logically complex and high-quality nature of this data enables the model to generalize to single-hop QA tasks not encountered during the instruction tuning phase, further confirming the reliability of our synthetic data. Detailed procedures are available in Appendix C.

### 4.5 Scaling Analysis

**Data Scaling Analysis**   To evaluate how the size of high-quality data affects model performance, we experiment on LLaMA3-8B (Dubey et al., 2024) by varying the training data volume. The results, depicted in Figure 14, illustrate a clear relationship between the amount of data and the performance. As the dataset size increases, model performance adjusts accordingly, demonstrating the significance of high-quality data scaling in enhancing the model efficacy.

**Hop Scaling Analysis**   To assess the impact of multi-hop data on model performance, we increased the number of hops in the dataset while keeping the training data volume constant. This approach isolated the effect of multi-hop reasoning on model outcomes. As indicated in Figure 15, there is a clear positive correlation between the number of hops and model performance. The data demonstrate that with more hops, the model achieves higher accuracy and robustness. These results demonstrate the effectiveness of using high-quality multi-hop data to enhance the model's capability for complex reasoning tasks.

## 5 Related Work

Recent efforts have aimed to enhance the performance of LLMs in handling longer contexts. LongLLaMA (Xiong et al., 2023) demonstrates the impact of incorporating long text data during various pre-training stages. LLaMA2-80K (Fu et al., 2024b) highlights the significance of using a domain-balanced, upsampled long text corpus to improve long text capabilities, requiring only a 5B-token corpus for effective comprehension. ICLM (Shi et al., 2024) enhances long-text reasoning by transforming pre-training data into knowledge graphs and splicing adjacent documents. To improve the model's ability to follow long text instructions, LongAlpaca (Chen et al., 2024b) combines a 9K paper question-answering (QA) corpus with 3K short instruction samples. In contrast, LongAlign (Bai et al., 2024a) utilizes Claude (Anthropic, 2023) to produce 10K QA pairs for training. Additionally, ChatQA (Liu et al., 2024b) enhances long-context QA performance by incorporating manually annotated data. Building on these approaches, ChatQA2 (Xu et al., 2024a) further incorporate existing long-text datasets, such as Natural Questions (NQ) (Kwiatkowski et al., 2019).

The method closest dataset is Quest (Gao et al., 2024), which constructs QA pairs from document data and splices documents based on QA pair correlations, resulting in a close-sourced single-hop QA corpus. In contrast, our approach models document correlations first, then create multi-hop QA pairs using related intra-document data. Additionally, we offer systematic analysis, open-source datasets, and significantly improved models.

## 6 Conclusion

In conclusion, our proposed Multi-agent Interactive Multi-hop Generation (`MIMG`) framework, which includes a quality verification agent, a single-hop question generation agent, a multiple question sampling strategy, and a multi-hop question merger agent, achieves high-quality, diverse instruction data. Our experiments show that this synthetic data notably enhances performance, even surpassing models trained on larger human-annotated data, highlighting the effectiveness of our approaches.

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

APPENDIX

# A  DATA CONSTRUCTION DETAILS

The construction of long-text multi-hop question-and-answer datasets is based on a structured approach leveraging pre-trained document corpora. This section outlines the methodology used for data collection, processing, and validation across multiple domains and languages.

## A.1  SOURCE DATA OVERVIEW

The primary source of long-text data is a pre-trained document corpus that spans nine distinct domains. The corpus includes data from both Chinese and English sources, ensuring a comprehensive multilingual dataset. The domains covered are:

- **Books (eBooks)**: A collection of various eBook formats that provide diverse literary content. Academic Papers: Scholarly articles sourced from repositories such as arXiv and CNKI. These datasets reflect cutting-edge research across multiple disciplines.

- **Finance**: Data from financial documents and discussions, including the ChatGLM-fin dataset, which encompasses various financial reports and conversational data related to financial analysis.

- **Knowledge**: Information extracted from online encyclopedic sources, including Baike-Wiki and Pile-Wikipedia, covering a broad range of general knowledge.

- **Science**: Data from reputable scientific sources, including Kepuchina and ScienceDaily, that focus on advancements in various scientific fields.

- **Law**: Legal documents and case law from the Pile-Freelaw dataset, providing insight into legal precedents and interpretations.

- **Medicine**: Medical literature, including publications from Pile-PubMed Central, which includes peer-reviewed medical research and case studies.

- **Technology**: Content derived from technical discussions and knowledge-sharing platforms such as Pile-StackExchange.

- **Web Resources**: Web data extracted from open-source platforms, specifically the Pile-OpenWebText2 dataset, reflecting general web-based information.

Each domain was selected to ensure the inclusion of diverse, domain-specific content that could support the generation of robust and accurate multi-hop question-and-answer sequences. A more fine-grained analysis can be seen in Figure 13 (a).

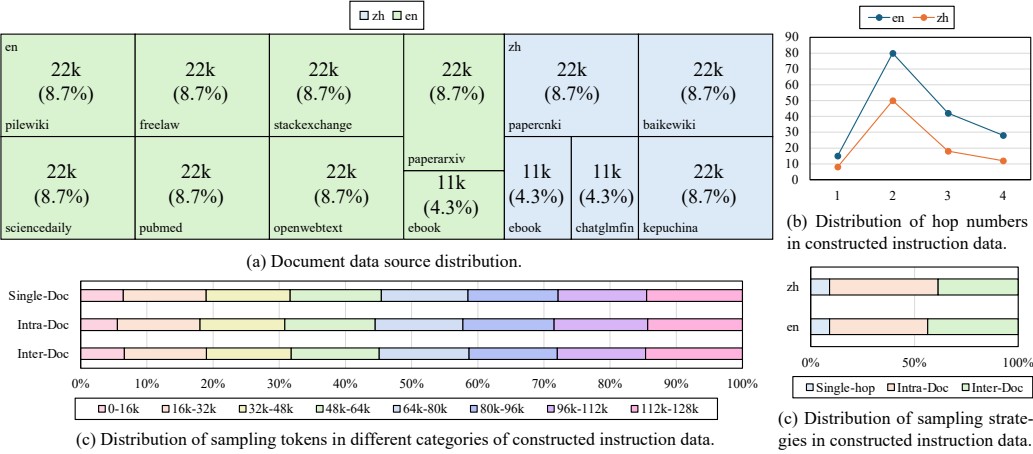

Figure 13: The analysis of constructed dataset distribution.

Additionally, inspired by Kim et al. (2023) and Chen et al. (2024a), CoT has the ability to bring powerful performance improvements to the instruction tuning. What's more, as shown in Figure 17, after adding CoT, the performance of the model has indeed improved. Therefore, in all our data synthesis processes, the answer contains a reasoning path. Furthermore, since LLMs often cannot fit all the document information that is extremely long documents, we perform truncation segmentation on the documents input to the model. After generating the sample, refill the document with other documents to a fixed length.

## A.2 MULTI-HOP QUESTION AND ANSWER DATA CONSTRUCTION

The construction of multi-hop question-and-answer datasets involved a rigorous process to ensure both linguistic accuracy and domain relevance. The methodology is as follows:

### A.2.1 DATASET CURATION

For each domain, data was independently curated to maintain a clear distinction between different knowledge sources. This allows for more focused and accurate multi-hop questions that are relevant to the particular field of study.

### A.2.2 QUALITY VERIFICATION AGENT

The first module in our framework is Quality Verification Agent, which ensures that the generated questions and answers meet a certain standard of quality. We use InternLM2-20B (Cai et al., 2024) as the backbone and set the quality score threshold to 8.5. Moreover, the prompts are as follows:

> Suppose you are a professional annotator, and you need to annotate the generated questions, rationales, and answers according to the context. Specifically, your tasks are as follows:
>
> - First, determine whether the questions and answers are in documents provided in context.
>
> - Then, you need to determine whether the problem is a multi-hop problem, using multi-hop logic.
>
> - At the same time, you need to judge whether the question conforms to commonsense logic. Does the question conform to common sense in a normal context? Is the logic smooth?
>
> - In addition, you need to rate the overall data quality from three aspects: logical rationality and fluency, question complexity, and answer clarity. All scores are between 0 and 10.
>
> - Before giving an annotation, you need to give your rationale.
>
> [[DOCUMENTS]]
> {chunk}
> [[QUESTION]]
> {question}
> [[ANSWER]]
> {answer}
>
> Finally, you should give me an overall quality mark in the format:
> '''{"in_document": BOOL, "domain_similarity": NUMBER, "quality": NUMBER}'''

### A.2.3 SINGLE-HOP QUESTION GENERATION AGENT

The Single-hop Question Generation Agent is responsible for generating fundamental single-hop questions, which are characterized by their simplicity and directness.

In this framework, we employ Qwen2-72B-Instruct (Yang et al., 2024) as the foundational model, utilizing it to synthesize data through a question-answering paradigm. The process begins with the

generation of prompts designed specifically for question creation, initiating a structured approach to the formulation of these queries.

---

The document content is as follows:

{chunk}

Extract the questions contained in the above document, and the extracted questions should meet the following conditions:

- No pictorial information should be included in the extracted questions;

- No referential information should be included in the extracted questions;

- Ensure the completeness of the extracted questions; if they are multiple-choice questions, provide corresponding option information, remove line breaks, and place the question body in a single question;

- If the document contains concepts such as numbers, time, people, or places, questions that involve this information must be extracted;

- The extracted questions should be presented in a parseable list format, such as ["xxx", "xxx"]. If there are no valuable questions, output an empty list [];

- Try to extract as many valuable questions as possible, but do not include duplicate questions;

- Extract no more than three questions;

Extracted questions:

---

Based on the questions extracted, the prompt for answer generation is as follows:

---

Generate answers to a given series of questions based on the content of the document, which must meet the following conditions:

- Respond based on the content in the document;

- If there is no corresponding answer to the question in the document, please reply based on your own knowledge;

- If the question is about factual issues such as numbers, time, people, places, etc., please provide the answer directly, and different question and answer pairs should be distinguished by line breaks;

The document content is as follows:

{chunk}

The problems are as follows:

{question}

The corresponding answers are as follows:

---

### A.2.4 MULTIPLE QUESTION SAMPLING

This strategy further enhances the generation of multi-hop instructions by selecting questions that address diverse elements within the document. This method facilitates the creation of comprehensive, multi-hop, long-text question-answer datasets that are meticulously customized to reflect the characteristics and requirements of specific domain data sources. The organization of the relevant documents begins by embedding them into vectors, where BGE-zh-1.5 and BGE-en-1.5 (Xiao et al., 2023) models are used to map the documents into 768-dimensional vectors. Following the methods inspired by Shi et al. (2024), the document vectors are embedded using Faiss to facilitate storage and efficient retrieval. This process relies on measuring vector distances to retrieve the 10 nearest documents for each document, creating a document graph.

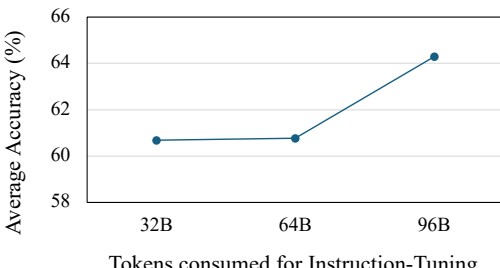
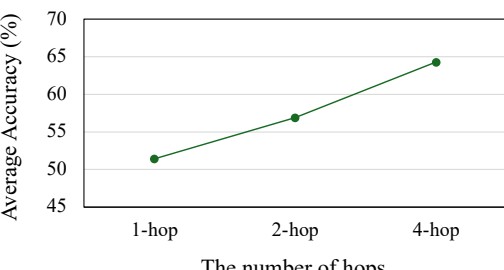

Figure 14: Analysis of the impact of different training dataset sizes on the average accuracy score.

Figure 15: Analysis of the impact of hop on model performance, where 1-hop is the reproduced version of the Quest (Gao et al., 2024) dataset.

Subsequently, a circular search strategy is employed to generate paths that consist of multiple documents, with the maximum path length constrained to 20. This process continues until all documents are sampled, with these paths serving as the initial sets of multiple related documents.

After conducting a sampling analysis, we observed the hop distribution in the constructed data, as illustrated in Figure 13 (b). Additionally, the distribution corresponding to the sampling strategy is depicted in Figure 13 (c).

### A.2.5 MULTI-HOP QUESTION MERGING AGENT

Multi-hop questions are designed to require reasoning across multiple data points, either within a single domain or spanning different domains. This approach ensures that responses cannot be derived from isolated facts; rather, they necessitate a more profound comprehension and integration of the dataset's overall content.

To achieve this, the Multi-hop Question Merging Agent consolidates single-hop questions into well-structured multi-hop queries. This process demands information synthesis from various sections of the document, promoting a deeper level of understanding and engagement. For the model architecture, we employ Qwen2-72B-Instruct (Yang et al., 2024) as the base model. The specific prompt for merging two QA pairs is as follows:

---

Based on the given two question-answer pairs, synthesize up to one question answer pair that matches the real scenario. The synthesized question-answer pair should meet the following conditions:

- If both questions and answers are time-related, a comparative question can be synthesized to compare the order in which two events occur;

- If both questions and answers are related to the character, it can be synthesized to determine which character better fits the description of the composite question;

- The synthesized answer should provide the corresponding reasoning process, and the synthesized answer should make as much use of the content in the given two answers as possible;

- Do not arbitrarily change the original information of two questions and answers;

- The generated questions and answers are strictly output in JSON format using {"question": xxx, "answer": xxx}. Synthesized question-answer pairs should not have any line breaks;

The correct answers to two questions are as follows:
{qa1}
{qa2}
The synthesized question-answer pair is:

---

## B  HIGHEST QUALITY STRATEGIES DETAILS

To achieve the highest quality data, we deliberately prioritize the use of GPT-4o as the backbone for all processes, fully disregarding cost constraints. This decision is driven by the understanding that ensuring the best data quality is paramount for the success of our project. Furthermore, to maintain and enhance performance during the exploration phase, we implement a comprehensive range of strategies aimed at maximizing the data retention rate.

Specifically, for the Quality Verification Agent, we employ a multi-faceted approach that includes more-perspectives scoring mechanisms, the addition of rationales, the integration of multiple perspectives, and the application of detailed guidelines. For the Single-hop Question Generation Agent, we have adopted a question-then-answer strategy. This approach is complemented by the incorporation of rationales, which provide context and justification for each query generated. Additionally, we require LLMs to generate only one question per query, which is intended to reduce the logical burden on the model, thereby improving the coherence and relevance of the questions produced. In the case of Multiple Question Sampling, we utilize BGE embeddings for the retrieval of questions. This technique is applied both within individual documents (intra-document) and across multiple documents (inter-document). Finally, for the Multi-hop Question Merging Agent, we employ a strategy that involves merging questions and answers using document references. This method ensures that the merged questions and answers are contextually aligned and coherent. Notably, we have opted to remove the rationale for merging in this process, as we found that it adds unnecessary complexity without significantly improving the quality of the merged content.

## C  INSTRUCTION TUNING EXPERIMENTS DETAILS

### C.1  TRAINING DETAILS

All models were trained using 64 A800*80G GPUs with the DeepSpeed+ZeRO-1 framework. The maximum sequence length was set from 4K to 128K, with any sequences exceeding this length truncated from the right. The training process utilized the Adam optimizer with a learning rate of $3 \times 10^{-5}$, $\beta_1 = 0.9$, and $\beta_2 = 0.95$.

To enhance training efficiency, we employed a packing strategy that concatenates training samples to reach the maximum sequence length. Additionally, Flash Attention (Dao et al., 2022; Dao, 2024) is used to accelerate the computation of the attention mechanism. The global batch size consisted of 4 million tokens, and the entire dataset is trained over one epoch.

### C.2  EVALUATION DETAILS

Based on the methodology proposed by Bai et al. (2024a), evaluating Token F1 using a model optimized through Chain of Thought (CoT) (Wei et al., 2022) reasoning proves to be challenging.

| | InterLM-2.5-7B-Enhance | | | | | InterLM-2.5-7B-Enhance + `LongMIT` | | | | |
|---|---|---|---|---|---|---|---|---|---|---|
| | 4k | 8k | 16k | 32k | 128k | 4k | 8k | 16k | 32k | 128k |
| S-NIAH Subtask-1 | 99.00 | 99.00 | 100.00 | 100.00 | 16.00 | 99.00 | 99.00 | 99.00 | 99.00 | 97.00 |
| S-NIAH Subtask-2 | 100.00 | 99.00 | 100.00 | 99.00 | 97.00 | 100.00 | 100.00 | 100.00 | 100.00 | 100.00 |
| S-NIAH Subtask-3 | 99.00 | 98.00 | 99.00 | 99.00 | 100.00 | 99.00 | 99.00 | 99.00 | 99.00 | 100.00 |
| MK-NIAH Subtask-1 | 97.00 | 98.00 | 97.00 | 88.00 | 58.00 | 100.00 | 100.00 | 98.00 | 99.00 | 90.00 |
| MK-NIAH Subtask-2 | 99.00 | 99.00 | 96.00 | 81.00 | 28.00 | 99.00 | 100.00 | 100.00 | 95.00 | 63.00 |
| MK-NIAH Subtask-3 | 95.00 | 90.00 | 56.00 | 14.00 | 0.00 | 96.00 | 91.00 | 70.00 | 33.00 | 2.00 |
| MV-NIAH | 99.25 | 99.50 | 99.50 | 94.50 | 84.50 | 99.00 | 99.00 | 97.00 | 93.50 | 89.50 |
| MQ-NIAH | 98.00 | 98.75 | 97.50 | 94.00 | 86.00 | 100.00 | 100.00 | 100.00 | 99.25 | 94.25 |
| VT | 91.20 | 91.80 | 98.60 | 97.40 | 0.00 | 96.60 | 97.80 | 98.80 | 95.60 | 94.20 |
| FWE | 85.33 | 87.00 | 84.67 | 91.00 | 71.67 | 86.00 | 89.00 | 86.67 | 90.67 | 78.33 |
| CWE | 83.40 | 67.10 | 34.70 | 26.40 | 0.10 | 75.60 | 40.40 | 8.50 | 6.10 | 0.20 |
| Single Hop QA | 90.00 | 80.00 | 81.00 | 75.00 | 42.00 | 92.00 | 84.00 | 82.00 | 80.00 | 58.00 |
| Multi Hop QA | 70.00 | 67.00 | 64.00 | 53.00 | 35.00 | 73.00 | 72.00 | 68.00 | 63.00 | 47.00 |
| Average | 92.78 | 90.32 | 85.23 | 77.87 | 47.56 | **93.48** | **90.09** | **85.15** | **81.01** | **70.27** |

Table 2: The evaluation performance on Ruler (Hsieh et al., 2024) benchmark based on `LongMIT` extended to 128K.

---

**Context**

...

Passage 6: **Francisco Giner de los Ríos**

Francisco Giner de los Ríos (10 October 1839 in Ronda, Spain – 18 February 1915 in Madrid) was a philosopher, educator and one of the most influential Spanish intellectuals at the end of the 19th and the beginning of the 20th century...

Passage 20: **Taifa of Ronda**

...The taifa was ruled by a family from the Berber Banu Ifran tribe of North Africa. Its capital was the city of Ronda. From 1065 until 1091, the taifa was under the control of the Taifa of Seville, led by Abbad II al-Mu'tadid.

---

| **Question** | **Golden Answer** |
|---|---|
| In which year did Taifa of Francisco Giner de los Rios' birthplace cease to exist? | **1065** |

---

**Answer in LLaMA3-8B trained on LongMIT**

Reasoning process: The birthplace of Francisco Giner de los Rios, Ronda, Spain, ceased to exist in 1065. This is because the Taifa of Ronda, which was centered in Ronda, Spain, and existed from 1039 to 1065, was conquered by the Taifa of Seville, led by Abbad II al-Mu'tadid, in 1065.

Final conclusion: The Taifa of Francisco Giner de los Rios' birthplace ceased to exist in **1065**.

---

**Answer in LLaMA3-8B trained on LongAlign**

The passage states that Mission San Francisco Solano officially ceased to exist on November 3, 1834, when it was designated a First Class Parish. So Taifa of Francisco Giner de los Rios' birthplace ceased to exist in **1834**.

Figure 16: The case study of the effectiveness of `LongMIT`.

To address this limitation, we employ GPT-4 as a consistency evaluator. Our testing demonstrates that the error rate of GPT-4 in this role remains consistently low, with deviations falling within a 2% margin. The corresponding prompt used is outlined below:

> Suppose you are a professional annotator. Given the result predicted by a model, you need to annotate whether the "[[PREDICTION]]" is consistent with the given "[[REFERENCE]]" based on the "[[QUESTION]]".
> [[QUESTION]]
> {question}
> [[PREDICTION]]
> {predictions}
> [[REFERENCE]]
> {answer}
> Finally, you should give me an annotation in the format:
> "' { "short_pred_answer": "xxx", "predict_consistency": BOOL } "'

## D  CASE STUDY

To gain a more nuanced and intuitive qualitative understanding of our model's performance, we conducted a detailed case study, resulting in two significant findings:

- **Impact of Instruction Quality:** As illustrated in Figure 17, models trained with high-quality multi-hop instruction data, specifically the `LongMIT` dataset, exhibit enhanced logical reasoning capabilities. These models are better equipped to process and analyze extensive textual information, enabling them to derive more accurate and reliable reasoning. In contrast, models trained using traditional, lower-quality instruction data, such as `LongAlign` (Bai et al., 2024a), demonstrate a reduced capacity for logical reasoning. This comparison underscores the importance of the quality of training data in developing models that can effectively handle complex reasoning tasks, especially when dealing with long and intricate texts.

---

**Context**

...

Passage 6: **Francisco Giner de los Ríos**

Francisco Giner de los Ríos (10 October 1839 in Ronda, Spain – 18 February 1915 in Madrid) was a philosopher, educator and one of the most influential Spanish intellectuals at the end of the 19th and the beginning of the 20th century…

Passage 20: **Taifa of Ronda**

The Taifa of Ronda (Arabic: طائفة رندة) was a medieval Berber taifa kingdom centered in Moorish al-Andalus in what is now southern Spain. It existed from 1039 to 1065. The taifa was ruled by a family from the Berber Banu Ifran tribe of North Africa. Its capital was the city of Ronda. From 1065 until 1091, the taifa was under the control of the Taifa of Seville, led by Abbad II al-Mu'tadid.

---

**Question** | **Golden Answer**

In which year did Taifa of Francisco Giner de los Rios' birthplace cease to exist? | **1065**

---

**Answer with CoT**

Reasoning process: The birthplace of Francisco Giner de los Rios, Ronda, Spain, ceased to exist in 1065. This is because the Taifa of Ronda, which was centered in Ronda, Spain, and existed from 1039 to 1065, was conquered by the Taifa of Seville, led by Abbad II al-Mu'tadid, in 1065.

Final conclusion: The Taifa of Francisco Giner de los Rios' birthplace ceased to exist in **1065**.

---

**Answer w/o CoT**

**1976.**

Figure 17: The case study of whether utilize reasoning process for instruction tuning.

- **Role of Rationale Incorporation in Training:** Furthermore, as depicted in Figure 16, our analysis reveals that the inclusion of additional rationales during the training process significantly enhances the model's ability to focus on relevant information within long texts and make precise inferences. This finding is particularly evident when comparing models that underwent Chain-of-Thought (CoT) (Wei et al., 2022) training with those that did not. Specifically, models that lacked CoT training tend to falter during inference, often generating erroneous outputs, such as the completely incorrect answer "1976". On the other hand, models that were fine-tuned with CoT training not only demonstrate a coherent logical reasoning process but also consistently arrive at the correct answer, "1065". This result highlights the critical role of rationale-based training in improving the model's reasoning accuracy and its ability to tackle complex inferential challenges.

