# OpenReview forum: "What are the Essential Factors in Crafting Effective Long Context Multi-Hop Instruction Datasets? Insights and Best Practices"
_ICLR.cc/2025/Conference — Submitted to ICLR 2025_

### Official Review · Reviewer_LHkC · 2024-11-02

**Soundness:** 3
**Presentation:** 3
**Contribution:** 2
**Rating:** 6
**Confidence:** 3

**Summary:**

This paper introduces a Multi-agent Interactive Multi-hop Generation (MIMG) framework designed to enhance the quality of multi-hop instruction data for long-context tasks. The framework includes four main components: a Quality Verification Agent, a Single-hop Question Generation Agent, a Multiple Question Sampling Strategy, and a Multi-hop Question Merging Agent. Through these components, the proposed MIMG framework significantly improves the quality, diversity, and relevance of synthetic instruction data, which surpasses performance metrics achieved by models trained on larger human-annotated datasets.

**Strengths:**

The main strengths of this paper include:

(1). Innovative Multi-agent Generation Framework: The proposed Multi-agent Interactive Multi-hop Generation (MIMG) framework incorporates multiple agents (Quality Verification Agent, Single-hop Question Generation Agent, Multiple Question Sampling Strategy, and Multi-hop Question Merging Agent), significantly improving the quality and diversity of generated data.

(2). Extensive Experimental Validation: The paper systematically investigates various document selection, question merging, and validation strategies, backed by experiments across multiple models and domains, demonstrating the practical effectiveness and generalizability of the framework.

(3). Enhanced Model Performance: Models trained with MIMG-generated data show an average improvement of 7.54% over those trained with larger, human-annotated datasets, underscoring the framework’s value in boosting long-context capabilities in large language models.

**Weaknesses:**

The main limitations of this paper are:

1). The primary weakness of this paper lies in its limited novelty. The contributions primarily emphasize engineering implementations and optimizations rather than presenting groundbreaking theoretical or methodological advancements. While the proposed framework demonstrates effective improvements in long-context, multi-hop instruction datasets, it largely builds upon existing concepts and technologies in a structured engineering fashion.

2). Limited Analysis of Long-term Effects on Model Robustness: While the paper demonstrates improvements in performance, it lacks a detailed investigation into how the synthetic multi-hop data affects model robustness and generalizability over long-term use, particularly in non-training contexts.

3). Potential Bias in Synthetic Data Quality Verification: The quality verification process, although effective, relies on automated scoring and classification from LLMs. This approach may introduce bias, particularly in complex, nuanced cases where human judgment could differ, impacting the interpretability and reliability of the data.

4). Token Cost of Rationale-based Generation: While rationale-based question generation can enhance quality, the paper notes that it significantly increases token consumption, raising concerns about its efficiency and scalability in resource-constrained environments.

5). Minimal Exploration of Alternative Frameworks: The study primarily focuses on the MIMG framework without thorough comparisons to alternative data synthesis or augmentation frameworks, limiting insights into how it performs relative to other potential approaches.

**Questions:**

1.  Lack of Analysis on Failure Cases: There is limited discussion on the types of tasks or data where the proposed method may underperform. An analysis of failure cases or limitations in specific scenarios would provide a more balanced view of the framework's practical utility.

2. A notable contradiction in this paper is the claim that "stronger LLMs can generate better single-hop questions" While the proposed framework aims to improve data generation quality and efficiency, the reported performance gains do not appear to match those achieved by simply using a stronger LLM. This inconsistency raises questions about the practical benefits of the proposed method, especially considering its added complexity. If a straightforward upgrade to a more powerful LLM yields comparable or superior results, the value of implementing this multi-agent framework diminishes. This aspect weakens the paper's argument for the proposed method as a more effective solution than alternative, less complex approaches.

3. To facilitate readers’ understanding of the related work in this field, it would be more effective to place the "Related Work" section immediately after the "Introduction." Currently, this section is written in a very general and unstructured manner, which makes it challenging to follow. Structuring the "Related Work" section into specific subcategories—such as "Large Language Models (LLMs)," "Multi-hop Instruction Datasets," etc.—would improve readability and provide a clearer context for the presented work.

---

> ### Author Response · Authors · 2024-11-20
>
> **Question 1:** Discussion about the novelty of this article.
>
> **Answer 1:** Thanks for your thoughtful advice. We would like to make it clear to you that our innovation is not focused on engineering optimization. In fact, we have brought a lot of insights that have not been discovered before. For example, the large model is not a good long context annotator but a better long context selector. In addition, we also analyze the shortcomings of quality classification on long contexts and also analyze the corresponding long context biases.
>
> Moreover, we first pay attention to the importance of multi-hop long context synthesis data for long context training, and provided a set of general long text data synthesis frameworks and strategies available to the academic community to increase the long text performance of the model.
>
> **Question 2:** Automated scoring and classification of LLMs may introduce bias, especially in complex, nuanced situations where human judgment may differ, thereby affecting the interpretability and reliability of the data.
>
> **Answer 2:** Thank you for your detailed feedback. In fact, this is the flaw of all LLM-based automated processes. To this end, we have chosen models and strategies that are as consistent with humans as possible, which precision score reaches 96.43%.
>
> **Question 3:** This paper points out that it significantly increases token consumption, raising concerns about its efficiency and scalability in resource-constrained environments.
>
> **Answer 3:** Thanks for your constructive comments. The core consumption of tokens for long text data synthesis is still long context input, and the output of CoT actually only accounts for 1/10 of the total tokens. We don’t think this will bring too much additional resource consumption.
>
> **Question 4:** The study focuses primarily on the MIMG framework without conducting a thorough comparison with other data synthesis or augmentation frameworks.
>
> **Answer 4:** Thanks for your valuable suggestion. In fact, our focus is on the long text field, and the current data in this field are based on Self-Instruct. There is not even a systematic framework. In fact, some existing long text work is basically a subset of our MIMG. For example, the prompt of Self-Instruction is almost the same as that of Single-hop Question Generation Agent. Self-Instruction + LLM recheck is also consistent with the structure in which we add Quality Verification Agent. Hence, we think we have a thorough comparison to a certain extent.
>
> **Question 5:** An analysis of failure cases or limitations in specific scenarios needed to be considered to provide a more balanced view of the usefulness of the framework.
>
> **Answer 5:** Thank you for your constructive comments. We found that a smaller part of the results will lead to over-reasoning, and it may be necessary to combine other single-hop reasoning situations for more practical expansion in the future. We will conduct more case analysis and discussion in the next version.
>
> **Question 6:** The performance gains in this paper appear to be comparable to those achieved by simply using a more powerful LLM. This inconsistency raises questions about the practical benefits of the proposed approach, especially given its added complexity.
>
> **Answer 6:** Thanks for your thoughtful suggestions. We want to clarify that the conclusion linking superior models to enhanced performance is primarily supported by the application of our MIMG framework, rather than by the mere comparison between baseline models and MIMG. Furthermore, as shown in Figure 12, the quality of the data synthesized using Qwen2 within the MIMG framework significantly exceeds that generated by GPT-4 using the Self-Instruct strategy. This illustrates the effectiveness of our framework, even more effective than backbone replacement. Furthermore, as illustrated in Table 1, our approach consistently outperforms prior efforts utilizing GPT-3.5, GPT-4, and even human-created data, which demonstrates and further proves the effectiveness of our framework.
> In addition, GPT4 is often impractical for training data synthesis. It is also important to note that the large-scale training data, distilled from only 4 billion tokens using GPT-4, incurs a cost exceeding $10,000, rendering such an approach impractical for many applications.
>
> **Question 7:** Suggestions for improving the readability of related work.
>
> **Answer 7:** Thank you for your detailed reply. I will adjust the content of the related work part in detail in the next version.

---

> > ### Comment · Reviewer_LHkC · 2024-12-02
> >
> > Thanks to the author for the reply, which solved most of my concerns. I also read the comments of other reviewers and I will improve my score.

---

> ### Author Response · Authors · 2024-11-28
>
> Dear Reviewer LHkC,
>
> As the discussion deadline is extended, we are actively looking forward to your further feedback. Thanks for your effort and understanding!
>
> Kindest regards,
>
> Authors of ICLR Submission 8639

---

> > ### Author Response · Authors · 2024-12-02
> >
> > Dear Reviewer LHkC,
> >
> > As there is **only 1 day remaining**, we kindly request your feedback to confirm that our response  effectively address your concerns.
> >
> > If there are any remaining issues, we would greatly appreciate the opportunity to address them to ensure the quality of our work. We sincerely hope that **you find our response convincing. Please consider revisiting your rating.**
> >
> > Kindest regards,
> >
> > Authors of ICLR Submission 8639

---

### Official Review · Reviewer_ztgo · 2024-11-03

**Soundness:** 3
**Presentation:** 3
**Contribution:** 3
**Rating:** 6
**Confidence:** 4

**Summary:**

This paper primarily focuses on synthesizing multi-hop, high-quality instruction data. The authors propose a data generation framework that incorporates multiple components, including a quality verification agent, a single-hop question generation agent, a multiple question sampling strategy, and a multi-hop question merging agent. Through experimental analysis, the authors identify the most effective strategies for each component and combine them to produce the final synthesized long-context instruction data. The experiments demonstrate that these synthesized data can enhance model performance.

**Strengths:**

1. Compared to previous multi-hop data generation methods like Self-Instruct, the MIMG framework significantly enhances the proportion of multi-hop data, as well as the diversity and quality of the data.
2. The authors conduct a thorough analysis of various potentially impactful strategies, such as document selection strategies and the impact of question merging methods. This provides practical references for future research endeavors.
3. The synthesized long context dataset (LongMIT) effectively enhances long-context utilization in experiments.

**Weaknesses:**

1. Although the author provides a detailed analysis of the impact of different strategies on the multi-hop data ratio, quality, or diversity in various components, they do not analyze **the impact of these components on the final performance**. Specifically, the roles of the Quality Verification Agent, Single-hop Question Generation Agent, Multiple Question Sampling, and Multi-hop Question Merger Agent in the final framework are not discussed. Analyzing these would help demonstrate the independent contributions and practical necessity of each module.
2. Although the author compares the cost tokens of the proposed method in Section 4.2, Figure 12 still shows that LongMIT-GPT4o has **more than four times the cost tokens compared to Self-Instruct-GPT4o**. Considering that the method introduces multiple agents and complex merging strategies, this significantly increases the computational resources required while improving model performance, which may affect the feasibility of practical applications.

**Questions:**

1. When analyzing different strategies, the author uses metrics such as retention ratio and average score. Could you provide a more detailed description and implementation method for these metrics to help readers better understand?

---

> ### Author Response · Authors · 2024-11-20
>
> **Question 1:** Analyzing these will help demonstrate the independent contribution and practical necessity of each module.
>
> **Answer 1:** Thanks for your thoughtful suggestions. To address your concerns, we analyzed and evaluated individual components of the agent architecture on human-annotated data quality. As illustrated in Table 1, each component contributes positively to overall data quality. Specifically, the Multi-hop Question Merger Agent significantly enhances the multi-hop quality of the dataset, while the Multiple Question Sampling mechanism improves the diversity of generated data. Moreover, the Single-hop Question Generation Agent plays a pivotal role in balancing both the quality and diversity of the data. Finally, the Quality Verification Agent serves as a foundational mechanism, ensuring a lower bound for quality and further enhancing the dataset’s overall standard.
>
> |  | High-quality | Diversity | Multi-hop |
> |:--|:--:|:--:|:--:|
> | Baseline | 61.3 | 53.4 | 33.1 |
> | MIMG | 94.8 | 88.2 | 94.8 |
> | w/o Multi-hop Question Merger Agent | 85.0 | 85.7 | 45.7 |
> | w/o Multiple Question Sampling | 92.8 | 65.4 | 89.4 |
> | w/o Single-hop Question Generation Agent | 82.4 | 69.4 | 73.2 |
> | w/o Quality Verification Agent | 78.3 | 88.1 | 91.0 |
> ||
>
> Table 1: Results of the ablation study on MIMG, reporting the data quality score as the average of three independent manual evaluations.
>
> **Question 2:** Figure 12 shows that the cost token of LongMIT-GPT4o is more than four times that of Self-Instruct GPT4o, which may affect the feasibility of practical applications.
>
> **Answer 2:** Thank you for your constructive comments. I think there is a misunderstanding here.
> First of all, the annotation of LongMIT-GPT4o in our figure is not accurate, which means that MIMG are equipped with all strategies that consume tokens but have higher data quality.
> Additionally, as shown in Figure 12, we would like to clarify that assuming that an equal amount of high-quality data is to be generated, the average token consumption of our method is about 5.3k, while the token consumption of using GPT-4o+Self-intstruct is close to 6.6k.
>
> **Question 3:** Can you give a more detailed description and implementation method for indicators such as retention rate and average score to help readers better understand?
>
> **Answer 3:** Thank you for your constructive comments. I greatly agree with your point of view. Our retention rate refers to the proportion of data that will be retained after being filtered by an LLM through a certain threshold. This is based on our observation that LLMs are better selectors rather than better annotators. LLMs have higher retention rates and are more consistent with humans.
>
> Additionally, the average score refers to the average score of the model. Statistically speaking, the higher the score, the higher the quality.

---

> ### Comment · Reviewer_ztgo · 2024-11-23
>
> Thank you for your response, but I still have the following questions:
>
> 1. Regarding Question 1, I still have some concerns. For the three metrics - high-quality, diversity, and multi-hop, there doesn't seem to be clear evidence in the paper demonstrating their impact on instruction-tuning performance. Could the authors provide evidence of how each module in MIMG affects final instruction-tuning performance, or provide direct evidence (including the magnitude) of how these three metrics impact instruction-tuning performance?
> 2. Regarding Question 2, I still don't understand your explanation. Could you provide accurate annotations on the figure regarding LongMIT-GPT4o and Self-Instruct-GPT4o?
> 3. Regarding Question 3, Thank you for your explanation, so does your retention rate refer to the data filtered by the Quality Verification Agent? Could you provide specific details such as the threshold value and the criteria for determining the threshold? Also, could you provide specific details about how the model determines the "average score"?

---

> > ### Author Response · Authors · 2024-11-24
> >
> > **Question 1:** For the three metrics - high-quality, diversity, and multi-hop, there doesn't seem to be clear evidence in the paper demonstrating their impact on instruction-tuning performance. Could the authors provide evidence of how each module in MIMG affects final instruction-tuning performance, or provide direct evidence (including the magnitude) of how these three metrics impact instruction-tuning performance?
> >
> > **Answer 1:** Thank you for your detailed response. To address your concerns, we conduct further sampling and annotation of several instruction-tuning dataset. The annotation quality is summarized in Table 1. Based on a comparison with Table 1 in the original paper, we draw the following conclusions:
> > 1. High-quality annotations significantly enhance model performance in foundational long-context comprehension tasks, such as those in DuReader.
> > 2. Diversity in annotations plays a crucial role in improving the model's overall performance across various tasks.
> > 3. The Multi-hop property is particularly impactful for complex reasoning tasks requiring integration of multiple long context clues, as exemplified by datasets like HotpotQA and MusiQue.
> >
> > |  | High-quality | Diversity | Multi-hop |
> > |:--|:--:|:--:|:--:|
> > | NQ | 90.2 | 63.2 | 10.3 |
> > | ChatQA2 | 83.1 | 80.3 | 30.4 |
> > | LongAlign | 87.7 | 83.8 | 52.6 |
> > | LongAlpaca | 70.3 | 87.9 | 50.7 |
> > | MIMG | 94.8 | 88.2 | 94.8 |
> > ||
> >
> > Table 1: Results of the data quality of different instruction datasets, reporting the data quality score as the average of three independent manual evaluations.
> >
> > **Question 2:** Regarding Question 2, I still don't understand your explanation. Could you provide accurate annotations on the figure regarding LongMIT-GPT4o and Self-Instruct-GPT4o?
> >
> > **Answer 2:**
> > - **Original Annotation:** LongMIT-GPT4o  $\rightarrow$  **Revised Annotation:** LongMIT-Best-Strategy.
> > 	- Add all strategies that yield better performance, but may incurring higher cost. These include using GPT-4o as the backbone model, incorporating additional rationales, merging questions with corresponding documents, and other related techniques.
> > - **Annotation:** Self-Instruct-GPT4o
> > 	- This strategy involves prompting GPT-4o to autonomously generate questions and corresponding answers based on the provided document, leveraging its self-instruction capabilities for required outputs.
> >
> > **Question 3:** Regarding Question 3, Thank you for your explanation, so does your retention rate refer to the data filtered by the Quality Verification Agent? Could you provide specific details such as the threshold value and the criteria for determining the threshold? Also, could you provide specific details about how the model determines the "average score"?
> >
> > **Answer 3:** Thank you for your thoughtful reply. Our detailed explanation is as follows:
> > - Yes. The retention rate refers to the proportion of data that remains after being filtered by the Quality Verification Agent relative to the original dataset.
> > - The threshold value we use is 8.5. This value was determined based on an internally annotated verification set comprising 200 items. The threshold corresponding to the highest precision on this set was selected as the standard parameter for subsequent evaluations.
> > - Regarding the "average score," our scoring strategy assigns a quality score ranging from 0 to 10 for each data point. We evaluate the entire dataset, collect the quality scores, and calculate the average score by taking the

---

> > > ### Comment · Reviewer_ztgo · 2024-11-25
> > >
> > > 1. Thank you very much for your explanation! However, the new results still don't establish a convincing connection between these three metrics and final performance. For example, we can see that NQ's high-quality is far better than ChatQA2, LongAlign, and LongAlpaca, but looking at the final performance, other methods all perform better than NQ. Does this suggest that the high-quality metric isn't important?Additionally, LongAlpaca's Diversity is better than LongAlign's, but the actual final performance is worse than LongAlign's. Does this suggest that diversity isn't playing a role in this case?
> > > 2. Thank you for your explanation. I understand the difference. Perhaps corresponding modifications could be made in the PDF to improve the article's clarity.
> > > 3. Thank you for your explanation. Looking forward to you providing further precision results for other thresholds on the annotated verification set.
> > > 4. Regarding the "average score," could you provide some previous works or corresponding experimental results showing that these "quality scores" are truly effective? Additionally, the paper has similar omissions of many key experimental details, which severely impacts readers' understanding of the paper.

---

> > > > ### Author Response · Authors · 2024-11-25
> > > >
> > > > **Question 1:** Thank you very much for your explanation! However, the new results still don't establish a convincing connection between these three metrics and final performance. For example, we can see that NQ's high-quality is far better than ChatQA2, LongAlign, and LongAlpaca, but looking at the final performance, other methods all perform better than NQ. Does this suggest that the high-quality metric isn't important?Additionally, LongAlpaca's Diversity is better than LongAlign's, but the actual final performance is worse than LongAlign's. Does this suggest that diversity isn't playing a role in this case?
> > > >
> > > > **Answer 1:** Thank you for your quick reply. However, as we described earlier, we should not just look at the Average Acc Score. By this, we may never observe anything valuable. The observed results require the consideration of multiple factors and the specific requirements of different tasks:
> > > >
> > > > **MAIN CONCLUSION: High quality underpins fundamental long-context understanding, and high diversity promotes out-of-domain generalization.**
> > > >
> > > > **OBSERVATION:**
> > > > 1. NQ exhibits high-quality data, effectively improve its utility to in-domain tasks, such as DuReader. On the other hand, its deficiency in diversity reduces its effectiveness in out-of-domain scenarios, thereby lowering its overall average accuracy.
> > > > 2. Conversely, while LongAlign demonstrates higher diversity, its quality is comparatively insufficient. This compromises its fundamental comprehension abilities, as evidenced by its significantly weaker performance on **basic** tasks like DuReader, ultimately leading to suboptimal overall results.
> > > >
> > > > In short, we believe that quality assurance addresses the basic capabilities of long texts, while diversity expands long context capabilities in more fields based on basic capabilities.
> > > >
> > > > **Question 2:** Details about thresholds.
> > > >
> > > > **Answer 2:** Thank you for your approval. We will discuss this issue further in next version.
> > > >
> > > > **Question 3:** Regarding the "average score," could you provide some previous works or corresponding experimental results showing that these "quality scores" are truly effective? Additionally, the paper has similar omissions of many key experimental details, which severely impacts readers' understanding of the paper.
> > > >
> > > > **Answer 3:** Thank you for your detailed review. In fact, our work has analyzed the consistency between quality score and human assessments, as illustrated in Figure 3. Specifically. Specifically, Figure 3(a) demonstrates that the Kappa coefficient for the agreement between the scoring mechanism and human evaluators exceeds 0.50, significantly surpassing the performance of classification strategies that directly label items as high or low quality. Furthermore, Figure 3(b) shows that the scoring mechanism achieves a precision level of 96.43% relative to human evaluators, underscoring its effectiveness as a robust data screening strategy.
> > > >
> > > > Additionally, we acknowledge the value of discussing prior research in this area. And we are also happy to share other papers on scoring: Several studies have shown that large-scale models exhibit a certain degree of alignment with human judgment when generating quality scores [1-3]. Notably, their performance extends to specialized domains, such as medicine, where these models demonstrate reasonable consistency with expert evaluations [4].
> > > >
> > > > We will add more experimental details in the next version to deepen the researchers’ understanding.
> > > >
> > > > [1] Chang et al. A Survey on Evaluation of Large Language Models. ACM Trans. Intell. Syst. Technol.
> > > >
> > > > [2] Chen et al. Exploring the use of large language models for reference-free text quality evaluation: A preliminary empirical study. IJCNLI 2023 Findings.
> > > >
> > > > [3] Lee et al. Unleashing Large Language Models’ Proficiency in Zero-shot Essay Scoring. EMNLP 2024 Findings.
> > > >
> > > > [4] Helm et al. AI in the ED: Assessing the efficacy of GPT models vs. physicians in medical score calculation. The American Journal of Emergency Medicine.

---

### Official Review · Reviewer_or1P · 2024-11-03

**Soundness:** 2
**Presentation:** 3
**Contribution:** 2
**Rating:** 5
**Confidence:** 4

**Summary:**

This paper proposes the Multi-agent Interactive Multi-hop Generation (MIMG) framework, which incorporates a Quality Verification Agent, a Single-hop Question Generation Agent, a Multiple Question Sampling Strategy, and a Multi-hop Question Merger Agent. This framework enhances data quality, with over 85% of the data being high-quality, multi-hop, and diverse.

**Strengths:**

1. The motivation of this paper is clear.
2. The exploration of methods within each agent module of the framework is thorough.

**Weaknesses:**

1. The paper contains some errors; for example, Figure 10 shows only one image but is labeled (a).
2. While the authors have explored methods within each agent module of the proposed framework to enhance data generation quality, there is a lack of ablation studies between the agents, making it unclear which agent contributes the most.
3. The experiments are not sufficiently generalized, as they were only evaluated on InternLM. I believe validation on widely used models like the LLaMA series is necessary.
4. The experimental comparisons in the paper are somewhat confusing: it is unclear whether the authors aim to propose a SOTA dataset or a framework for generating data. If it is the latter, I believe comparisons with other works that generate multi-hop data using the same LLM should be included.

**Questions:**

1. The evaluation criteria for the data need further clarification, especially for metrics like Diversity.
2. In the experimental comparisons within the Data Utilization section, I am a bit confused about the details of LongMIT’s experimental data, such as the number of samples, the number of tokens, and comparisons with other datasets.
3. Please refer to the questions mentioned in the Weaknesses.

---

> ### Author Response · Authors · 2024-11-20
>
> **Question 1:** Some minor annotations and grammatical errors in this paper.
>
> **Answer 1:** Thank you for your detailed suggestions, we will correct the corresponding minor errors point by point.
>
> **Question 2:** Ablation studies between agents are lacking, so it is unclear which agent contributes the most.
>
> **Answer 2:** Thank you for your insightful suggestions. To address your concerns, we analyze the contributions of various components of the agent on human-annotated data quality. As summarized in Table 1, each component positively contributes to the overall quality. Notably, the Multi-hop Question Merger Agent significantly enhances the multi-hop quality of the dataset, while the Multiple Question Sampling mechanism effectively increases the diversity of the generated data. Furthermore, the Single-hop Question Generation Agent plays a crucial role in simultaneously improving both the quality and diversity of the data. Additionally, the Quality Verification Agent serves as a critical safeguard, maintaining a lower bound to ensure model quality and thereby further enhancing the data's overall integrity.
>
> |  | High-quality | Diversity | Multi-hop |
> |:--|:--:|:--:|:--:|
> | Baseline | 61.3 | 53.4 | 33.1 |
> | MIMG | 94.8 | 88.2 | 94.8 |
> | w/o Multi-hop Question Merger Agent | 85.0 | 85.7 | 45.7 |
> | w/o Multiple Question Sampling | 92.8 | 65.4 | 89.4 |
> | w/o Single-hop Question Generation Agent | 82.4 | 69.4 | 73.2 |
> | w/o Quality Verification Agent | 78.3 | 88.1 | 91.0 |
> ||
>
> Table 1: Results of the ablation study on MIMG, reporting the data quality score as the average of three independent manual evaluations.
>
> **Question 3:** These experiments were only evaluated on InternLM, while widely used models such as the LLaMA series is necessary to be evaluated.
>
> **Answer 3:** Thank you for your insightful advice. We want to claim that our work does not only conduct experiments on InternLM. All our analytical experiments are validated on multiple LLMs, including Qwen2, InternLM2, Gemini-1.5, GPT-4o-mini and GPT-4o. In addition, in terms of model training, as shown in Table 1, we apply the experiments on both InternLM2 and LLAMA-3.
>
> **Question 4:** It is unclear whether the authors aimed to present a SOTA dataset or a framework for generating the data.
>
> **Answer 4:** Thanks for your constructive response. In fact, both these two points are our contribution. The logic of our paper is to explore a solution with a higher quality-cost ratio, and then scale a better dataset based on this solution to optimize the model. Therefore, we discuss the optimal solution for each part and gradually optimize it as the best solution for the next step. So, at each step we discussed the quality of data generated by different models.
>
> Furthermore, our research primarily focuses on long-context tasks, an area where the synthesis of multi-hop data has not been systematically explored for long-text applications. To the best of our knowledge, we are the first to introduce this consideration in such tasks. Furthermore, as outlined in Table 1 of the paper, the datasets we compared with were predominantly synthesized using advanced models, including GPT-3.5-Turbo, GPT-4, and even human annotations, which much surpass the capabilities of our data synthesis backbone. Despite this, our synthesized datasets demonstrate superior performance compared to these benchmarks.
>
> We will add more discussions in the next version.
>
> **Question 5:** The evaluation criteria for data need to be further clarified, especially indicators such as diversity.
>
> **Answer 5:** Thank you for your constructive comments. I greatly agree with your point of view. Diversity refers to whether an annotator can find data with the same semantic meaning in previously annotated samples as a proportion of all data. We will discuss each indicator in detail in the next version.
>
> **Question 6:** Details of LongMIT data, including number of samples and number of tokens.
>
> **Answer 6:** Thanks for your valuable feedback. To address the problem, we conduct a detailed statistical analysis of sample size and token consumption across various datasets. As shown in Table 2, while our token count is comparable to that of NQ, our dataset contains significantly fewer samples and outperforms NQ by more than 10% in terms of performance. Interestingly, although NQ includes a much larger number of samples than most other datasets, its performance is inferior even to datasets with smaller sample sizes, such as LongAlpaca and LongAlign.
>
> | Dataset Name | Sample Size | Token |
> | :-- | :--: | :--: |
> | LongAlpaca | 12.00k | 0.11B |
> | LongAlign | 9.89k | 0.17B |
> | ChatQA2 | 128.00k | 1.22B |
> | NQ | 315.20k | 4.56B |
> | LongMIT | 64.40k | 5.07B |
> ||
>
> Table 2: The volume statistics results of different datasets.

---

> > ### Author Response · Authors · 2024-11-28
> >
> > Dear Reviewer or1P,
> >
> > As the discussion deadline is extended, we are actively looking forward to your further feedback. Thanks for your effort and understanding!
> >
> > Kindest regards,
> >
> > Authors of ICLR Submission 8639

---

> > > ### Author Response · Authors · 2024-12-02
> > >
> > > Dear Reviewer or1P,
> > >
> > > As there is **only 1 day remaining**, we kindly request your feedback to confirm that our response  effectively address your concerns.
> > >
> > > If there are any remaining issues, we would greatly appreciate the opportunity to address them to ensure the quality of our work. We sincerely hope that **you find our response convincing. Please consider revisiting your rating.**
> > >
> > > Kindest regards,
> > >
> > > Authors of ICLR Submission 8639

---

### Meta-Review · Area_Chair_5wvY · 2024-12-23

**Metareview:**

The paper introduces the Multi-agent Interactive Multi-hop Generation (MIMG) framework to synthesize high-quality, diverse multi-hop instruction datasets for long-context tasks. MIMG incorporates agents for quality verification, single-hop question generation, multiple question sampling, and multi-hop question merging. The framework reportedly achieves over 85% high-quality data and demonstrates improvements on selected benchmarks compared to human-annotated datasets.

While the problem is relevant, the contributions are incremental. The framework largely integrates existing methods with limited novelty, and the evaluation lacks sufficient baselines and ablation studies to isolate contributions of individual components. Key metrics, such as diversity and multi-hop performance, are poorly defined, and their impact on downstream tasks remains unclear. Additionally, the reliance on synthetic benchmarks and high token costs raises questions about scalability and generalizability.

Strengths include a structured approach to data synthesis and some empirical improvements. However, the lack of novelty, incomplete evaluations, and scalability concerns diminish the paper’s impact.

**Additional Comments On Reviewer Discussion:**

Reviewers noted strengths in addressing a critical area and improving synthetic datasets but raised concerns about limited innovation, unclear metrics, and evaluation rigor. The authors provided additional explanations and experiments during the rebuttal, but concerns about scalability, the practical utility of metrics, and insufficient comparisons to alternative frameworks remain unresolved. Further work is needed to address these gaps and validate the framework's broader applicability.

---

### Decision · Program_Chairs · 2025-01-22

Reject